# Cooperative Fe sites on transition metal (oxy)hydroxides drive high oxygen evolution activity in base

Yingqing Ou[1,2,7], Liam P. Twight[1,7], Bipasa Samanta[3,7], Lu Liu[1,4], Santu Biswas[3], Jessica L. Fehrs[1], Nicole A. Sagui[1], Javier Villalobos[5], Joaquín Morales-Santelices[5], Denis Antipin[5], Marcel Risch [5], Maytal Caspary Toroker [3,6] ✉ & Shannon W. Boettcher [1] ✉

Fe-containing transition-metal (oxy)hydroxides are highly active oxygen-evolution reaction (OER) electrocatalysts in alkaline media and ubiquitously form across many materials systems. The complexity and dynamics of the Fe sites within the (oxy)hydroxide have slowed understanding of how and where the Fe-based active sites form—information critical for designing catalysts and electrolytes with higher activity and stability. We show that where/how Fe species in the electrolyte incorporate into host Ni or Co (oxy)hydroxides depends on the electrochemical history and structural properties of the host material. Substantially less Fe is incorporated from Fe-spiked electrolyte into Ni (oxy)hydroxide at anodic potentials, past the nominally $Ni^{2+/3+}$ redox wave, compared to during potential cycling. The Fe adsorbed under constant anodic potentials leads to impressively high per-Fe OER turn-over frequency ($TOF_{Fe}$) of ~40 s$^{-1}$ at 350 mV overpotential which we attribute to under-coordinated "surface" Fe. By systematically controlling the concentration of surface Fe, we find $TOF_{Fe}$ increases linearly with the Fe concentration. This suggests a changing OER mechanism with increased Fe concentration, consistent with a mechanism involving cooperative Fe sites in $FeO_x$ clusters.

The electrolysis of water ($2H_2O \rightarrow 2H_2 + O_2$) to produce hydrogen fuel is critical for renewable-energy infrastructure[1,2]. Even in optimized electrolyzers, the efficiency is reduced by the slow kinetics of the oxygen evolution reaction (OER, $4OH^- \rightarrow 2H_2O + 4e^- + O_2$ in alkaline media)[3,4]. Among the many OER catalysts studied over decades, Fe is broadly pivotal in promoting the catalytic activity in alkaline media[5–7]. In particular, Ni and Co (oxy)hydroxides require Fe to achieve high activity, regardless of whether it is introduced intentionally or incidentally[8,9].

Sustaining the high activity of these Fe-containing (oxy)hydroxides requires soluble Fe species in the electrolyte due to the dynamic Fe exchange at the catalyst/electrolyte interface[10–13]. High-activity perovskite oxides also involve active Fe sites[14]. $La_{1-x}Sr_xCoO_3$ perovskites form $CoO_xH_y$ layers in alkaline electrolytes with enhanced activity in the presence of trace-level Fe electrolyte species[15]. $Ba_{0.5}Sr_{0.5}Co_{0.8}Fe_{0.2}O_3$ undergoes surface reconstruction and transforms to OER-active Co/Fe oxyhydroxides[16]. In fact, Fe-containing (oxy)hydroxides are ubiquitous

[1]Department of Chemistry and Biochemistry and the Oregon Center for Electrochemistry, University of Oregon, Eugene, Oregon 97403, USA. [2]School of Chemistry and Chemical Engineering, Chongqing University, 400044 Chongqing, China. [3]Department of Materials Science and Engineering, Technion—Israel Institute of Technology, Haifa 3200003, Israel. [4]School of Materials Science and Engineering, Chongqing University, 400044 Chongqing, China. [5]Nachwuchsgruppe Gestaltung des Sauerstoffentwicklungsmechanismus, Helmholtz-Zentrum Berlin für Materialien und Energie, Hahn-Meitner-Platz 1, 14109 Berlin, Germany. [6]The Nancy and Stephen Grand Technion Energy Program, Haifa, Israel. [7]These authors contributed equally: Yingqing Ou, Liam P. Twight, Bipasa Samanta. ✉e-mail: maytalc@technion.ac.il; swb@uoregon.edu

for OER-active materials including transition-metal oxides, sulfides, selenides, and phosphides[17–19].

Typically Ni-Fe and Co-Fe hydroxides adopt structures analogous to α-phase $Ni(OH)_2$ and $Co(OH)_2$, which consist of layers of $[M(OH)_6]$ octahedra with rotational disorder and water/ions intercalated into the interlayer space[20–23]. The $Fe^{3+}$ is thought to substitute for the $M^{2+}$ sites with the extra charge balanced by intercalated anions[24]. Prior to OER, Fe-doped α-$M(OH)_2$ is oxidized to (nominally) γ-MOOH, accompanied by a contraction of the M-O bond length and interlayer distance, consistent with stronger M-O bonds upon formal cation oxidation[25]. Fe substitution induces a positive shift of the apparent $M^{2+}/M^{3+}$ redox potential, indicative of electronic interactions between Fe and host Ni or Co cations. The oxidation state of Fe during OER at the active site remains a point of discussion. Both $Fe^{3+}$ and $Fe^{4+}$ were found under OER conditions for co-deposited Ni-Fe or Co-Fe (oxy)hydroxides, while in non-aqueous electrolyte $Fe^{6+}$ was identified and invoked as the active intermediate[26–29]. *Operando* Mössbauer spectra show the oxidation of $Fe^{3+}$ in Ni-Fe (oxy)hydroxide to $Fe^{4+}$ during OER and that these $Fe^{4+}$ species largely persist after the potential was decreased into a non-OER region[30]. Such $Fe^{4+}$ species were hypothesized to arise from fully coordinated internal sites within the NiOOH that were too kinetically slow to catalyze OER. In contrast, the detected $Fe^{4+}$ population in a $CoFeO_x$ film from a separate Mössbauer study correlated with OER activity, suggesting a central role of $Fe^{4+}$ in catalysis[31]. One hypothesis to explain this discrepancy would be the presence of different populations of Fe cations, for example in the interior regions of the (oxy) hydroxide sheets versus under-coordinated surface or edge Fe species.

We previously found evidence for different Fe local environments within the host (oxy)hydroxides affecting OER activity[11,32]. When cycled, both $NiO_xH_y$ and $CoO_xH_y$ adsorb $Fe^{3+}$ (intentionally added to the electrolyte) onto sites that were hypothesized to be easily accessible, i.e., at edges, corners, or defects in the two-dimensional (oxy)hydroxide structure. The Fe incorporated during the first voltammetry cycle dramatically enhanced the OER activity but had negligible influence on the host $NiO_xH_y$ redox-peak position or size, indicating weak electronic interaction between these "surface" Fe and the majority of the Ni cations. Repeated voltammetric cycling increased the amount of Fe incorporated, up to a Fe:Ni ratio of ~0.25, and the Ni redox wave shifted positive indicating strong coupling between the additionally added Fe and the Ni cations. Yet this additional Fe caused only a small increase in OER activity. These data led to the hypothesis that at least two general types of Fe sites exist in the (oxy)hydroxides, (1) OER-active surface sites from electrolyte Fe adsorption and (2) interior sites where Fe sits fully coordinated with bridging O(H) to neighboring M sites. Supporting this idea, $NiFeO_xH_y$ with proposed surface-attached FeOOH nanoclusters has been reported to be more OER-active than benchmark Ni-Fe catalysts[33,34]. The above findings illustrate that structural information or activity measurements for Fe-based sites collected from co-deposited (oxy)hydroxides are the weighted average of the multiple Fe environments which have very different properties. The precise intrinsic activity on the most active "surface" Fe, and the associated key OER mechanism, remain unknown.

Here we show how to confine the absorption of Fe to nominally surface sites where they have exceptionally high per-Fe activity yielding the catalysts Fe:$NiO_xH_y$ and Fe:$CoO_xH_y$. We study how the underlying structural features and dynamics of $NiO_xH_y$ and $CoO_xH_y$ affect the incorporation. Fe added into the electrolyte during chronoamperometry under OER conditions (positive of the $Ni^{2/3+}$ redox wave) yields electrolyte-adsorbed Fe species primarily on the surface (presumably at edge/defect sites) of Ni and Co (oxy)hydroxides. In contrast, potential cycling pure $NiO_xH_y$ in the presence of $Fe^{3+}(aq.)$ or $Co^{2+}(aq.)$ yields more-homogeneous Ni(Fe)$O_xH_y$ or Ni(Co)$O_xH_y$ phases, respectively. Similarly, we find that it is more difficult to incorporate $Fe^{3+}$ into the interior sites of $CoO_xH_y$ compared to $NiO_xH_y$, consistent

with the larger $Co^{2+}$-O(H) bond strength compared to $Ni^{2+}$-O(H)[35] and thus increased morphological stability of $CoO_xH_y$[36]. With this platform to control the location of the Fe-based active sites, we measure the intrinsic OER activity of surface Fe sites on Fe:$NiO_xH_y$ and Fe:$CoO_xH_y$ and compare these to activities with Fe at interior sites in $NiO_xH_y$. The turn-over frequency for OER, normalized to the number of Fe sites ($TOF_{Fe}$), increased linearly with the amount of adsorbed surface Fe until a saturation limit in sharp contrast to previous studies where the location of the Fe sites was uncontrolled[13]. We find an intrinsic $TOF_{Fe}$ of ~$40 s^{-1}$ at 350 mV for Fe:$NiO_xH_y$ which is at least five times higher than for benchmark co-deposited phases[7]. This finding illustrates a possible cooperative effect between multiple Fe sites on the surface of $NiO_xH_y$ and $CoO_xH_y$. Density-functional-theory (DFT) calculations suggest that this effect is derived from the ability of neighboring Fe atoms in a Fe-O-Fe cluster to share and stabilize positive charge during the oxidation of key intermediates, compared to a single $FeO_x$ supported on $NiO_xH_y$. These findings are important to understand and control structure in the design of higher-performance OER catalysts from earth-abundant metals for use in advanced electrolyzers[3,37] and photoelectrochemical systems.

## Results and discussion

### Mechanisms of foreign-cation incorporation into $NiO_xH_y$ and $CoO_xH_y$

We define two classes of sites where foreign ions can incorporate that contribute differently to the OER activity and electrochemical response. First, there are interior "bulk" sites where the foreign/incorporated cations are substituting either Ni or Co cations. These modify the electronic energies of the redox-active host metal atoms (Ni or Co) and thus influence the peak position of the Ni or Co redox waves during voltammetry[8,38,39]. Second, surface sites where the cations are adsorbed onto, rather than substituted into, the host $NiO_xH_y$ and $CoO_xH_y$ porous structures (Supplementary Fig. 1) where the coordination by water and terminal hydroxyls, as opposed to more-strongly bonded bridging ligands, makes them putative OER active sites[11,32]. These surface species have little effect on redox wave position, but generate most of the catalytic enhancement. To study these different sites, hydrated, electrolyte-permeable $NiO_xH_y$ and $CoO_xH_y$ films were cathodically electrodeposited from their associated nitrates (Supplementary Fig. 3) on Pt/Ti/glass substrates. Pt was used because it is relatively OER inactive even with in-situ formed $FeO_xH_y$ when Fe species are present in the electrolyte[40]. The porosity and thinness of the films ensured that almost all metal cations in the film are electrochemically active as confirmed by electrochemical microbalance studies[7,41]. SEM images of a representative $NiO_xH_y$ film (Supplementary Fig. 13) illustrate the porosity and roughness of the film and cross sections (Supplementary Fig. 14) suggest a thickness between 5 and 20 nm.

We use the redox response of Ni and Co cations to gain insight into the incorporation of foreign metal species from the electrolyte. Directly tracking the different Fe sites using electrochemistry is not possible as they do not provide a useful redox signature[42]; $Fe^{3+}$ oxidation occurs at potentials within the OER regime and $Fe^{3+}$ reduction occurs at potentials where the $NiO_xH_y$ and $CoO_xH_y$ host are electrically insulating. We expect that when foreign cations adsorb on the host, they should provide a distinct wave separate from that of the host (if they are redox active), while when incorporated substitutionally in interior sites they will exert an electronic effect manifesting as a shift of the host redox-wave potential.

When 0.1 ppm $Co^{2+}$ is added into the electrolyte during constant-potential polarization of $NiO_xH_y$ (Supplementary Fig. 5), we observed a new redox feature centered at 1.13 V vs. RHE (Fig. 1a) in the following voltammetry test, ~50 mV positive of the same wave for as-deposited $CoO_xH_y$ on Pt, which is thus assigned to $Co(OH)_2/CoOOH$. During the CA, the OER current increased by a factor of five after $Co^{2+}$ was added,

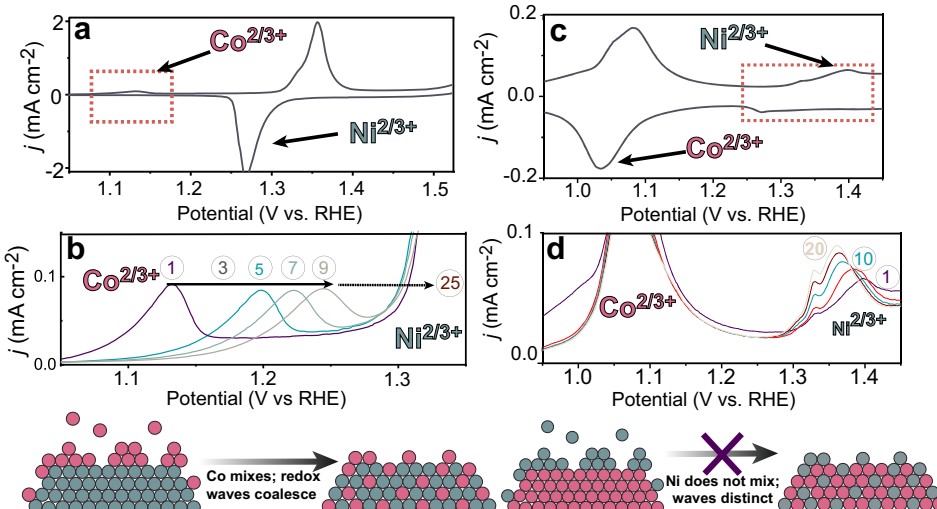

**Fig. 1 | Foreign cations interaction with oxyhydroxides via redox signatures.**
Cyclic voltammetry of Co-spiked $NiO_xH_y$ (**a**, **b**) and of Ni-spiked $CoO_xH_y$ (**c**, **d**) showing the evolution of redox features of the host metal hydroxide and phase formed by spiked ions under constant electrode potential. Inset numbers in (**b**) and (**d**) correspond to the cycle number, while the bottom cartoon illustrates schematically the process intentionally without atomic detail that is yet unknown. The data above was not $iR_u$-compensated.

consistent with $CoO_xH_y$, which is more OER active[38], absorbing on the $NiO_xH_y$.

With subsequent voltammetry cycles (Fig. 1b), the $CoO_xH_y$ wave shifts to higher potential, and the host $NiO_xH_y$ wave broadened (Supplementary Fig. 6). By 25 cycles, the characteristic $CoO_xH_y$ redox peaks have moved positive in potential and coalesced (Fig. 1b) with that of $NiO_xH_y$. The loss of the independent $CoO_xH_y$ redox peaks suggests that the Co cations initially adsorbed in a separate phase from the host $NiO_xH_y$ are now dispersed into the $NiO_xH_y$. The ability of $NiO_xH_y$ to accommodate Co cations in its bulk structure via voltammetry is verified by direct cycling $NiO_xH_y$ in $Co^{2+}$ containing electrolyte (Supplementary Fig. 8a). In this case, a ~10 mV negative shift of the $Ni(OH)_2$/NiOOH redox peaks are observed after 10 voltammetry cycles, while the $Co(OH)_2$/CoOOH signature wave is absent throughout the process. The negative shift of the Ni redox wave with $Co^{2+}$ incorporation is similar to that observed for co-deposited $Ni(Co)O_xH_y$ films and indicates strong electronic coupling between added-Co and host-Ni sites[43,44]. By analogy, similar processes are proposed when $NiO_xH_y$ is cycled in the presence of solution Fe cations that have no apparent redox signature.

After constant-potential polarization of $CoO_xH_y$ in $Ni^{2+}$-spiked KOH, subsequent voltammetry yields a wave centered at ~1.32 V vs. RHE (Fig. 1c), typical for the $Ni(OH)_2$/NiOOH couple. The OER current during CA for $CoO_xH_y$ is unaffected by spiking $Ni^{2+}$ into the electrolyte (Supplementary Fig. 7a), consistent with $NiO_xH_y$ being less OER-active than $CoO_xH_y$[38]. In contrast to the case of the $NiO_xH_y$ host, the new $Ni(OH)_2$/NiOOH wave that appears persists throughout the cycling process with only a slight negative shift of the peak potential (Fig. 1d), while the host $Co(OH)_2$/CoOOH wave position is unaffected (Supplementary Fig. 7b). The lack of electronic interaction between foreign Ni cations on the $CoO_xH_y$ host redox is consistent with Ni cations that persistently reside on the surface of $CoO_xH_y$ as a separate $NiO_xH_y$ phase despite cycling (Supplementary Fig. 8b). The inability of $CoO_xH_y$ to easily incorporate foreign cations, relative to $NiO_xH_y$, is consistent with its greater structural stability[45] and its stronger Co-O(H) bonds relative to Ni-O(H)[35,46]. Because the procedures used in these Co- and Ni-spiking experiments are identical to those used for Fe spiking in

activity measurements below, we propose that $Fe^{3+}$(aq.) adsorbs in a phase distinct from $CoO_xH_y$ and remains as such after cycling, but that $Fe^{3+}$ incorporates into the internal sites when the host $NiO_xH_y$ is electrochemically cycled. Evidence for separate phase formation by adsorbed electrolyte ions using this method has not been previously demonstrated and is critical to understanding the origin of exceptional $TOF_{Fe}$ described below.

The different locations of the foreign cations, in or on, $NiO_xH_y$ and $CoO_xH_y$ films appear driven by structural changes during voltammetry. Using an electrochemical quartz crystal microbalance (EQCM), the mass change of $NiO_xH_y$ and $CoO_xH_y$ films upon cycling was monitored during voltammetry (Supplementary Fig. 2). During the positive oxidative scan of $NiO_xH_y$, the film mass increases at the potential where $Ni(OH)_2$ oxidation is measured and continues into the OER region. The mass increase in the forward scan is ~0.33 μg·cm⁻², accounting for ~4.5% of total film mass (4.2 ± 0.2 g per mole $e^-$ passed). Mass loss is observed in the backward scan, particularly with the reduction of Ni species (~0.33 V vs. Hg/HgO). At the end of the cycle, the film mass returns to its initial value. For $CoO_xH_y$, a slight and irreversible mass gain (~0.2% of total film mass) is observed after the second voltammetry cycle, but no detectable mass change accompanies the oxidation of $Co(OH)_2$.

The mass gain with oxidation of $Ni^{2+}$ is thought to originate from $K^+$ and $OH^-$ intercalation along with the release of $H_2O$ in the interlayer space of $NiO_xH_y$[21,47,48]. Oxidation also causes contraction of the Ni-O bond and interlayer spacing, which introduces mechanical stress[21,25]. Restructuring of $Ni(OH)_2$ single-layer nanosheets into small nanoparticles was found by electrochemical atomic force microscopy (EC-AFM) suggesting that $Ni(OH)_2$ undergoes a dissolution-redeposition process during cycling[45,49]. When Fe or other cations exist in the electrolyte, these cations can exchange for Ni leading to atomic-level mixing with the host Ni. In contrast, the voltammetry of $CoO_xH_y$ does not involve obvious molecule/ion exchange based on EQCM and EC-AFM measurements and shows structural stability compared to $NiO_xH_y$[45], apparently making it difficult to incorporate foreign cations from the electrolyte into the structure interior.

## Confining Fe cations on the surface of $NiO_xH_y$ and $CoO_xH_y$

Using the insights from the Ni- and Co-spiking experiments, we add $Fe^{3+}$ during chronoamperometry to prepare $Fe:NiO_xH_y$ and $Fe:CoO_xH_y$ OER catalyst where Fe is surface-adsorbed. First, we applied ~1.55 V vs. RHE to $NiO_xH_y$ (nominally NiOOH at this potential) in Fe-free 1.0 M KOH for 3–5 min and recorded the (low) baseline OER activity. Then $Fe^{3+}$ was added to reach a concentration of 0.1 ppm (this low amount of Fe was selected to prevent bulk electrodeposition of nominally FeOOH we observed previously by electrochemical AFM at > 1 ppm[49]). A dramatic increase in OER current was immediately observed reaching a maximum after ~15 min suggesting the fast adsorption of Fe species on NiOOH, likely limited by mass transport of Fe species to the electrode. The incorporation of foreign cations at *fixed* potentials in the OER regime, and positive of the nominal $Ni^{2+/3+}$ redox wave, is important to limit the amount of host restructuring and associated inter-mixing that is driven by the redox transitions[45,49].

After ~15 min the OER current reaches a stable maximum of ~90× that of Fe-free NiOOH (without $iR_u$ compensation) and ICP-MS analysis of the catalyst film shows ~5 at. % Fe (relative to Ni). The same measurements were performed with $CoO_xH_y$ and the OER activity in Fe-free KOH was higher than $NiO_xH_y$, consistent with our previous reports[38]. Adding 0.1 ppm $Fe^{3+}$ in the electrolyte resulted in a ~7-fold increase in OER current (Fig. 2c), a much smaller enhancement than for $Fe:NiO_xH_y$. The observed activity difference between $Fe:NiO_xH_y$ and $Fe:CoO_xH_y$

likely derives from different intrinsic activities of surface Fe on the two chemically different hosts (see below).

The first voltammetry cycle of $Fe:NiO_xH_y$ after chronoamperometric (CA) Fe incorporation shows nearly identical redox response as Fe-free $NiO_xH_y$ (Fig. 2b), despite having dramatically enhanced OER activity. This implies that the Fe incorporated during CA is not yet electronically interacting with the majority of the host Ni metals. The $Fe:NiO_xH_y$ was then cycled 20 times in the Fe-spiked KOH (Fig. 2b, grey). The Ni wave shifts positively by 34 mV and the integrated peak area shrinks as the Fe incorporates into the oxyhydroxide structure to form $Ni(Fe)O_xH_y$ (that also contains surface-absorbed Fe) with almost no change in OER activity. The positive shift of the $NiO_xH_y$ wave is known to occur upon mixing with Fe by co-deposition[8]. This data implies an intrinsic activity difference for surface and internal Fe-based sites. Similarly, the first CV cycle of $Fe:CoO_xH_y$ after CA-spiking test shows almost the same redox peak position and area compared to $CoO_xH_y$, despite the enhanced OER activity (Fig. 2d). Subsequent voltammetry on $Fe:CoO_xH_y$ only slightly decreases the OER overpotential, with the redox waves almost unchanged. Bare Pt and Au substrates also absorb Fe increasing OER activity[40], but the Fe-spiked electrolyte barely affected the activity of control Pt electrodes under CA conditions here (Supplementary Fig. 4). Thus the enhanced activity observed for $NiO_xH_y$ or $CoO_xH_y$ loaded on Pt is derived from $Ni(Fe)O_xH_y$ or $Co(Fe)O_xH_y$, with the Fe-based active sites proposed to be all absorbed on the "surface" of the host oxyhydroxide.

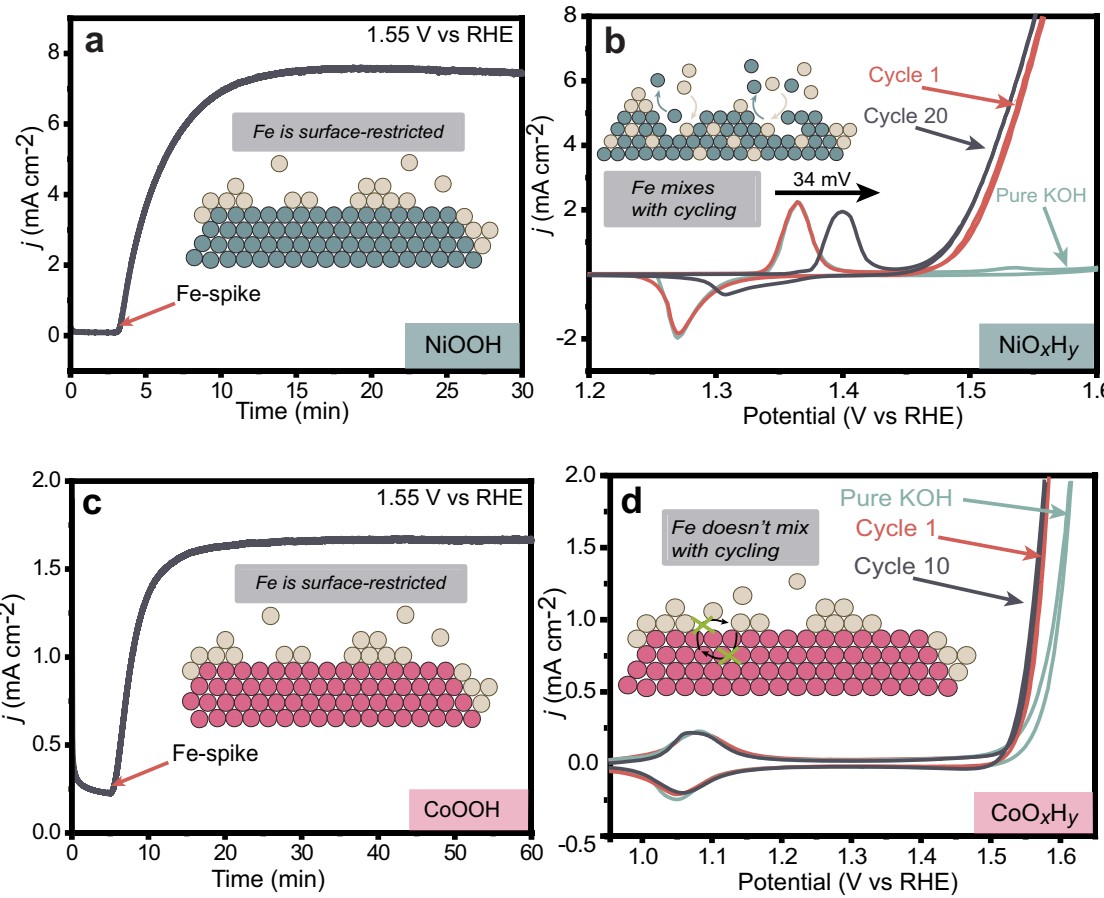

**Fig. 2 | Surface-confined Fe sites via chronoamperometric (CA) metal-ion-spiking.** CA measurements of **a** NiOOH and **c** CoOOH at 1.55 V vs. RHE. After starting the measurement in purified Fe-free 1.0 M KOH electrolyte, aqueous Fe(NO₃)₃ was added to a concentration of 0.1 ppm. The first voltammetry cycle (red, 10 mV/s) after Fe-spiking CA measurements shows the dramatic effect of Fe incorporation on the OER activities of $NiO_xH_y$ (**b**) and $CoO_xH_y$ (**d**), but a minimal

effect on the redox wave compared to the light green lines that show the initial voltammetry (cycle two) recorded in purified Fe-free 1.0 M KOH. The grey curves illustrate the large effect on the redox wave position for $NiO_xH_y$ but not for $CoO_xH_y$ after cycling, but that the OER activity does not further change much. The data is not $iR_u$ compensated. The cartoons illustrate schematically the process intentionally without atomic detail that is yet unknown.

## Intrinsic activity of Fe sites on metal (oxy)hydroxides

The OER turnover frequency (TOF) is defined as the total number of $O_2$ molecules generated per active site, per unit time[41]. One fundamental challenge is the identification and quantification of the true active sites to enable catalyst design. TOF for OER for these systems can be calculated in several ways[41]. The simplest is to use the total number of cations in the film (regardless of their location and chemistry, including Ni, Fe, Co, etc.) to calculate $TOF_{tm}$ which thus provides the average activity at all metal sites, simplifying the reality of many sites with a range of activities.

We calculate a TOF based on the total number of Fe cations ($TOF_{Fe}$), including both the interior and surface sites discussed here, as Fe is essential at the active site[50-52]. Our results show that Fe incorporates at surface sites at constant OER potentials and mixes into the interior only after cycling. The number of surface-adsorbed Fe sites can be controlled by stopping the CA experiment at different times while Fe is accumulating on the $NiO_xH_y$ or $CoO_xH_y$ (Supplementary Figs. 11, 12, 17 and 18), thus providing a route to create and study specific types of Fe sites in these important materials.

Figure 3a shows the $TOF_{Fe}$ values at $\eta = 300$ mV ($iR_u$ corrected) for Fe:NiOOH and Ni(Fe)OOH. For Fe:NiOOH, $TOF_{Fe}$ increases nearly linearly with the amount of Fe absorbed. Initially, the sample is -0.7 at.% Fe on $NiO_xH_y$ and the $TOF_{Fe}$ is $2.0 \pm 0.3$ s$^{-1}$. The $TOF_{Fe}$ increases until maximum Fe adsorption and OER current at 5.1 at.% Fe on $NiO_xH_y$ and the high $TOF_{Fe}$ of $10.4 \pm 1.4$ s$^{-1}$ at $\eta = 300$ mV is observed. This data

shows that each Fe-based site becomes more OER active as surface-absorbed Fe sites accumulate.

After reaching a maximum OER current at constant potential in Fe-spiked KOH, the resulting $Fe:NiO_xH_y$ was cycled (Supplementary Fig. 16) resulting in the absorption of additional Fe, including at internal sites, and thus a large decrease in $TOF_{Fe}$ as Fe increases to -20 at. % (Fig. 3a). These data are consistent with the hypothesis that the absorbed Fe at surface sites drives OER, while internal Fe sites are comparatively inactive. We also prepared mixed $Ni(Fe)O_xH_y$ via co-electrodeposition, for which Fe is homogeneously substituted for Ni. $TOF_{Fe}$ values for these co-deposited samples decrease with Fe content and are substantially smaller than those obtained from Fe spiking, consistent with an activity difference between surface and internal Fe sites. $TOF_{Fe}$ at $\eta = 350$ mV shows similar trends (Supplementary Fig. 19). The highest $TOF_{Fe}$ of ca. $40 \pm 2$ s$^{-1}$ at $\eta = 350$ mV is obtained at the maximum surface adsorption. Previous studies show that the co-deposited optimal $Ni_{0.75}Fe_{0.25}O_xH_y$ has a $TOF_{Fe}$ of ca. 9 s$^{-1}$ at $\eta = 350$ mV[38]. On a $TOF_{Fe}$ basis, the $Fe:NiO_xH_y$ reported here demonstrates remarkable activity among all alkaline OER catalysts[53].

The surface Fe sites also become more OER active with increasing number on the surface of $CoO_xH_y$ (Fig. 3b). At maximum adsorption, -2.4% Fe is adsorbed on $CoO_xH_y$ and the highest $TOF_{Fe}$ at $\eta = 300$ mV is $0.71 \pm 0.08$ s$^{-1}$. This value is smaller than the $TOF_{Fe}$ in $Fe:NiO_xH_y$ with similar absorbed surface Fe amount (-5 s$^{-1}$ at $\eta = 300$ mV with -2.5% Fe) showing a large intrinsic activity difference between Fe sites on $CoO_xH_y$ versus $NiO_xH_y$. The $TOF_{Fe}$ of co-electrodeposited $Co(Fe)O_xH_y$

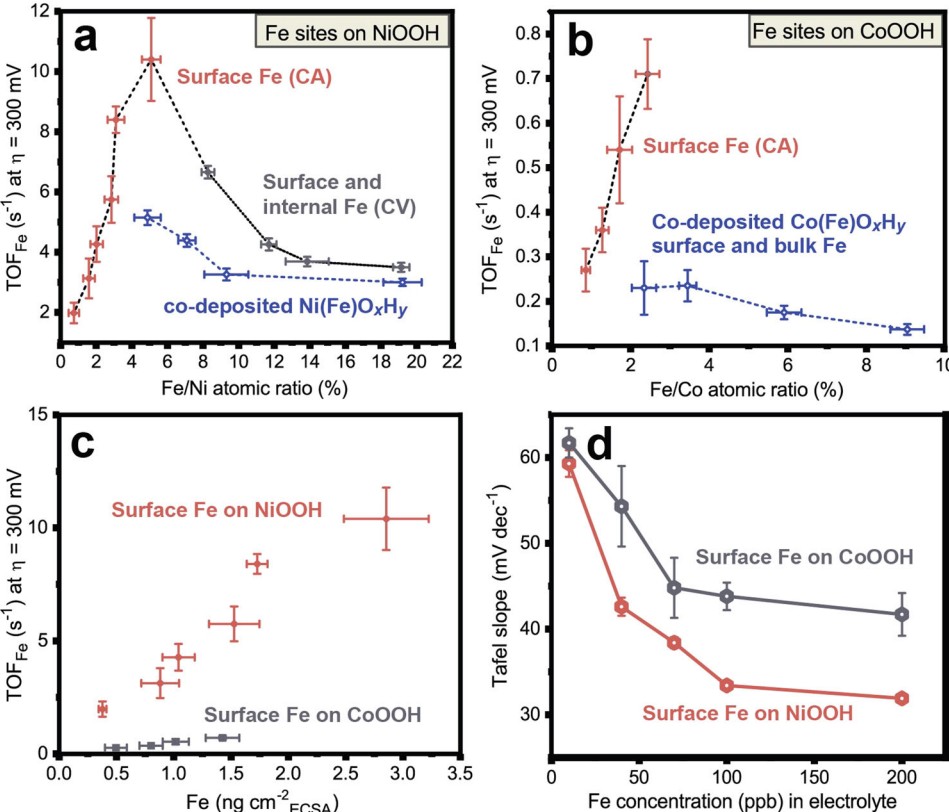

**Fig. 3 | Intrinsic OER activity measured by turnover frequency ($TOF_{Fe}$) at $\eta = 300$ mV.** The $TOF_{Fe}$ is calculated based on the mass of all Fe sites determined by ICP-MS of each dissolved film. **a** Correlation between $TOF_{Fe}$ and Fe/Ni atomic ratio for surface-confined Fe generated by CA, as well as mixed systems from cycling or co-deposition. **b** Correlation between $TOF_{Fe}$ and Fe/Co atomic ratio. **c** Correlation between the $TOF_{Fe}$ of surface-Fe sites on NiOOH (red) and CoOOH (grey) and the adsorbed Fe mass loading normalized by the electrochemical surface area of host oxyhydroxide. **d** Tafel slopes of $Fe:NiO_xH_y$ (red) and

$Fe:CoO_xH_y$ (grey) as a function of Fe concentration (10, 40, 70, 100, and 200 ppb) in 1.0 M KOH electrolyte. Tafel analysis was performed using constant current steps from 0.18 to 3.2 mA·cm$^{-2}$, with each step held for 3 min. The steps were then repeated in reverse order. Before Tafel analysis, constant potential OER in Fe-spiked electrolyte was performed until the maximum OER current was reached. All current values used for $TOF_{Fe}$ concentrations were $iR_u$-compensated where $R_u$ was $15.1 \pm 0.5$ Ω. Error bars represent one standard deviation from the average of triplicate measurements.

is, like the Ni system, also smaller than that of the surface Fe sites on $CoO_xH_y$ and decreases with Fe content (due to a larger fraction of less-active internal Fe sites). Because $NiO_xH_y$ and $CoO_xH_y$ films have different mass loading and electrochemically active surface area (ECSA), the $TOF_{Fe}$ values are shown as a function of mass of adsorbed Fe (from ICP-MS) normalized to geometric surface area (Supplementary Fig. 20) and ECSA (Fig. 3c). Under both metrics, the Fe sites on $NiO_xH_y$ exhibit higher $TOF_{Fe}$ than on $CoO_xH_y$, strong evidence for an intrinsic difference in site activity.

In contrast to our results, Chung and Markovic et al. observed a linear relationship between OER activity and the amount of absorbed Fe on several transition-metal (oxy)hydroxides including $NiO_xH_y$, $CoO_xH_y$, $Ni(Cu)O_xH_y$, and $Ni(Mn)O_xH_y$ by incorporation of Fe from electrolyte. This data was used to argue that each Fe site has similar activity and thus the improvement of OER catalysis mainly relies on increasing the number of Fe sites by increasing the absorption energy of the Fe on the host[13]. The difference between our work and this Markovic study is that the nature of the Fe sites was not previously controlled. The higher amount of Fe (> 18 atomic %) from their experiments indicates that Fe was not restricted to edge and defect sites and thus the number of active surface sites, versus fully coordinated interior ones, was unknown and uncontrolled and so the intrinsic activity of the two could not be separated nor cooperative effects between Fe sites discovered.

The above $TOF_{Fe}$ calculations demonstrate that the intrinsic OER activity of Fe sites is dependent on the local configuration: (1) surface Fe sites have much higher OER activity than bulk sites, (2) the intrinsic activity of surface Fe sites is affected by the interface with the host material, as exemplified by the difference in $TOF_{Fe}$ on $NiO_xH_y$ and $CoO_xH_y$ at a fixed Fe loading, and (3) the surface Fe sites show increasing activity on a per-site basis with the extent of surface Fe accumulation.

The emergence of surface-absorbed-Fe-site cooperativity can be explained simply. As more Fe is adsorbed, the likelihood of two or more Fe sites being located adjacent to each other increases. This may create a catalytic site with favorable electronics for adsorbate formation and evolution. There is precedent in the literature for clusters of different sizes to display different catalytic activity. For example, smaller clusters of $CoO_x$ were found to be more active for OER per Co site[54] and size-dependent catalysis by metal clusters is a well-known phenomenon which, in general, arises from optimal electronic structure, and hence absorption energies, at a particular size[55,56]. This, however, is but one possible explanation for the data, and other mechanisms invoking interfacial Ni-$FeO_x$ cooperation[33] could also be at play, especially considering that the intrinsic activity of Fe:$NiO_xH_y$ is greater than that of Fe:$CoO_xH_y$ at a comparable Fe loading. Related catalyst/support-type interactions of this kind are known for $FeO_xH_y$ thin films on Au substrates whose activity is much higher than when they are deposited on Pt[42].

## Electrokinetic analysis of cooperative Fe sites

The Tafel slopes of Fe:$NiO_xH_y$ and Fe:$CoO_xH_y$ decrease with increased Fe concentration in the electrolyte (Fig. 3d), which suggests Fe-absorption processes that modulates OER mechanism and not just number of active sites[10,13]. At low Fe concentration of 10 ppb, surface-Fe Fe:$NiO_xH_y$ and Fe:$CoO_xH_y$ have Tafel slopes of ~59 and ~62 mV dec$^{-1}$ respectively, suggesting a similar OER mechanism when small amounts of Fe are adsorbed, perhaps as isolated sites. Tafel slopes of ~33 and 32 mV dec$^{-1}$ are obtained for Fe:$NiO_xH_y$ 100 and 200 ppb Fe$^{3+}$ in the electrolyte (after the CA experiment until the maximum OER current results), like co-deposited $Ni_{0.75}Fe_{0.25}O_xH_y$. This result suggests that a similar mechanism is operative, likely due to the presence of cooperative surface Fe as the competent catalyst species in both. In comparison, (surface-Fe) Fe:$CoO_xH_y$ has Tafel slopes of ~44 and 42 mV dec$^{-1}$ with 100 and 200 ppb Fe$^{3+}$ in the electrolyte, slightly higher than those

for co-electrodeposited $Co_{1-x}Fe_xO_xH_y$ ($x = 0.33 \sim 0.79$ and Tafel slopes of 26–39 mV dec$^{-1}$) films[7]. In both Fe-free and 100-ppb-Fe electrolyte, Tafel slopes of Fe:NiOOH increase as a function of cycling (Supplementary Fig. 30) with the increase faster for the Fe-free electrolyte, consistent with both the desorption of surface Fe into the electrolyte for the Fe-free case and scrambling of the surface Fe with internal sites in the Fe-spiked electrolyte upon potential cycling. Further detailed electrokinetic analysis on a broader set of materials derived from these methods would be useful to understand how the proposed Fe-based local structures control apparent reaction pathways[57].

## Structural characterization

TEM with EDX mapping reveals the proposed Fe-oxo clusters must be small (Fig. 4 and Supplementary Figs. 14 and 15), likely of molecular dimensions, and consistent with electrochemical and ICP-MS data. Still, EDX is limited when low concentrations are being measured. We thus also quantified the amount of Fe incorporated in $NiO_xH_y$ during CA in 0.1 ppm Fe as a function of $NiO_xH_y$ mass loading (Fig. 4 and Supplementary Figs. 9 and 10) using ICP-MS to understand how Fe is incorporated. Both the integrated charge in the Ni redox wave and the double-layer capacitance ($C_{DL}$, measured in the oxidized state) of the films increased linearly with mass, showing the films are electrolyte-permeable and fully electron-accessible. The amount of Fe, ~10 atomic %, incorporated by 10 voltammetry cycles was constant with mass loading up to the maximum loading studied of ~100 nmol·cm$^{-2}$ Ni, consistent with cycling incorporating Fe throughout the $NiO_xH_y$. In contrast, the amount of incorporated Fe from constant-potential CA decreases from ~8 at. % to 3 at. % as the Ni loading is increased from 20 to 100 nmol·cm$^{-2}$. This is consistent with $FeO_x$ primarily absorbing on edge or defect sites. As $NiO_xH_y$ loading is increased, both new nanosheets form and existing nanosheets grow, thus decreasing the amount of edge-site-area per mass. Pair-distribution-analysis (PDF) also shows the size of coherent-scattering $Ni(Fe)O_xH_y$ domains increases with mass loading[58]. From the previous PDF data, we estimate that if all the edges of each coherently scattering domain, containing roughly 100–200 Ni cations, were decorated with a single row of absorbed $FeO_x$, this would correspond to roughly 20 at. % Fe. The 3–8 at. % observed here indicates that absorption of Fe is not uniform along all edges and that all the edges are not covered.

We next investigated the local structure of Fe adsorbed during CA on $NiO_xH_y$ using *operando* XAS (Supplementary Fig. 23). The Fe$^{3+}$ spiking experiments and EDX analysis suggest Fe species adsorbed during CA are at surface sites as low-nuclearity clusters of $FeO_xH_y$, which grow with time under positive polarization in the Fe-spiked solution. If so, we expect the Fe-O bond lengths of these species to more-resemble Fe oxyhydroxide and that subsequent cyclic voltammetry would lead to a contraction of the Fe-O bond due to incorporation into the NiOOH host (which has a shorter M-O bond length). This reasoning is based on the XAS measurements on co-deposited $NiFeO_xH_y$ by Bell and coworkers[25] where the Fe-O bond length measured under OER conditions had values that increased with the amount of Fe in the $NiFeO_xH_y$, with the pure $FeO_xH_y$ having the longest bonds. XAS measurements at the Fe K-edge were performed during polarization at 0.68 V vs Hg/HgO immediately after spiking with CA and again after cycling (Supplementary Fig. 25). The current increased immediately after Fe spiking and again after cycling (Supplementary Fig. 24), although these could not be normalized to the amount of Fe incorporated in this experiment. The similarity of the shape and position of the XANES edges of our sample and $Fe_2O_3$ corroborates that Fe is predominantly in a 3+ oxidation state (Supplementary Fig. 26); minor differences in shape may arise from difference in crystal structure. From the EXAFS data fitting (Supplementary Fig. 27 and Supplementary Table 2), we find that before cycling the Fe-O bond length was $1.953 \pm 0.004$ Å, and after cycling it contracted to $1.939 \pm 0.011$ Å; consistent with the hypothesis of Fe moving from Fe-

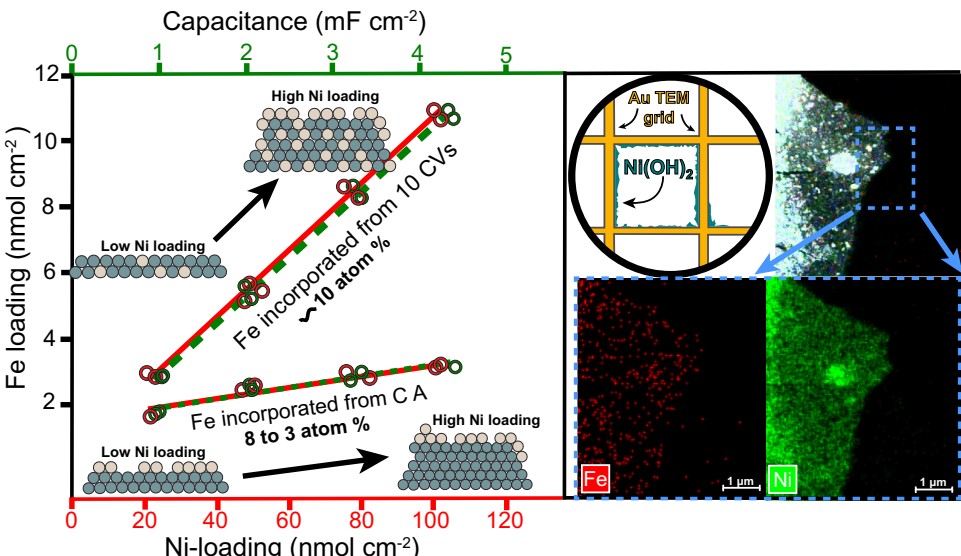

**Fig. 4 | Plot (left panel) of the change in Fe loading by incorporation during CA and with CA plus 10 subsequent CV cycles versus the total Ni mass loading and film capacitance.** All elemental data was obtained with ICP-MS measurement of the dissolved films. The inset depicts how the mol or at % changed for each Fe incorporation technique from the lowest to highest Ni loading. While Fe incorporates at roughly constant mol % when the Ni and Fe species are allowed to mix by CV, the mol % of Fe incorporated by CA decreases substantially as the Ni loading is increased. The right panel TEM-EDX images of $NiO_xH_y$ electrodeposited directly onto a gold TEM grid after Fe was adsorbed during chronoamperometry at 1.55 V vs RHE. While we observe localized Fe signal, it is not possible with this technique to distinguish between the surface-absorbed and internal/bulk sites nor to quantify accurately the size of the Fe clusters.

rich surface absorbed structures to internal sites. The determined coordination numbers (CN) of the first and second shell of near 5 for later stages of Fe adsorption are consistent with an Fe-oxo cluster of molecular dimensions larger than a dimer adsorbed on the surface (CN 3) but smaller than an extended crystal (CN 6), for which also further Fe-M shells should have been resolved. The error bounds on the EXAFS data, however, precludes a definitive conclusion from this XAS data alone, and further study with XAS and complementary *operando* techniques is needed to elucidate in detail the nature of the cooperative Fe sites formed by adsorption under OER conditions.

**Understanding OER mechanisms via DFT calculations**

To test how Fe cooperative interactions affect the OER mechanism and overpotential we built two model systems: (1) Fe cations separated and each coordinated by four hydroxides, one water, and one oxo bonded to the surface Ni denoted as "isolated Fe-O" species (Fig. 5a, top, note there are no Fe-O-Fe linkages), and (2) two adsorbed Fe cations directly bonded by a bridge oxygen ligand as a "Fe-O-Fe dimer" (Fig. 5b, top). These two models simulate the cooperation of neighboring surface-absorbed Fe-O species as a model active site, which may also be found on larger clusters. The structure of the adsorbed clusters at different view angles are shown in Supplementary Fig. 32. To build the surface, the bulk NiOOH structure was cleaved at the $0\bar{1}5$ plane due to its known activity[59–63]. The $0\bar{1}5$-oriented unit cell is then multiplied in 2×1×1 direction to have a Fe concentration of 9%, similar to the experimental values for maximum $TOF_{Fe}$. The bottom part of Fig. 5 shows elementary reactions steps and energy changes (eV) in the various mechanisms considered for the Fe-O and Fe-O-Fe modified surfaces. The theoretical overpotential ($\eta_{th}$) is defined as the voltage needed for all the reactions steps to have negative Gibbs free energies and are summarized in Fig. 5c and Supplementary Table 3.

Water oxidation at the axially coordinated water (*axial* pathway) is the common mechanism for both Fe-O-Fe dimers and isolated Fe-O species. Based on the calculated $\Delta G$ for each step in the *axial* mechanism (Supplementary Table 3), the Fe-O-Fe dimer has a $\eta_{th}$ of 0.50 V, while for isolated Fe-O it is 0.96 V. Critically, both the Fe cations in the Fe-O-Fe dimer change their oxidation state from 3 to 4 during

the step with the largest $\Delta G$ (Supplementary Table 6), while for the isolated Fe-O only case the Fe cation at the active site changes its oxidation state. To test whether the lower overpotential requires specifically two adjacent absorbed Fe cations, one of the Fe was replaced with Ni. In the Fe-O-Ni dimer, Fe changes its formal oxidation state from 4 to 5 while the Ni oxidation state is unchanged, yielding $\eta_{th} = 0.89$ V. For the isolated Fe-O species, Ni substitution at one of the Fe atoms reduces $\eta_{th}$ only from 0.96 V to 0.80 V. To check whether the lowered overpotential upon Fe dimerization is facet specific, the (001) surface of NiOOH (Supplementary Fig. 31) was also modeled[64]. Like the $(0\bar{1}5)$ surface, the Fe-O-Fe dimer on the (001) surface has roughly half the $\eta_{th}$ compared to the isolated Fe-O. The oxidation-state changes are similar as well with both $Fe^{3+}$ cations in the Fe-O-Fe dimer oxidized to $Fe^{4+}$ during the potential-determining step. In general, the investigation of the *axial* pathway for both types of surfaces showed that isolated monomers have higher OER overpotentials than dimers (when all surface-attached group atoms are Fe), in agreement with experimental $TOF_{Fe}$ trends. Beyond dimers, larger surface Fe-oxo clusters, as compatible with EXAFS analysis, may further stabilize the potential-determining intermediate by spreading oxidative charge over multiple Fe sites. We note that the actual experimental system likely includes various more-complex surface geometries with more absorbed Fe cations than the simple dimer considered in the calculation.

For the Fe-O-Fe dimer on the $0\bar{1}5$ surface other possible mechanisms were also investigated (Fig. 5b). The *axial* and *equatorial* pathways proceed through the same intermediates except that they take place at two different positions; both have similar $\eta_{th}$ of 0.50 and 0.47 V and in both cases the Fe oxidation state changes from 3 to 4. The *equatorial* pathway on the isolated Fe-O was also assessed and found impractical due to the instability of the water-coordinated structure. The *insertion* pathway shares the same first step that has the largest free-energy change with the *axial* pathway and thus both mechanisms have similar $\eta_{th}$. The key feature of the *bridge* mechanism that sets it apart from the others is that the bridge oxo group attached to two Fe and one Ni. When the bridging oxo group is deprotonated, all these surrounding metal atoms are oxidized; the two Fe and one Ni increase their oxidation state from +3 to +4 (Supplementary Table 6).

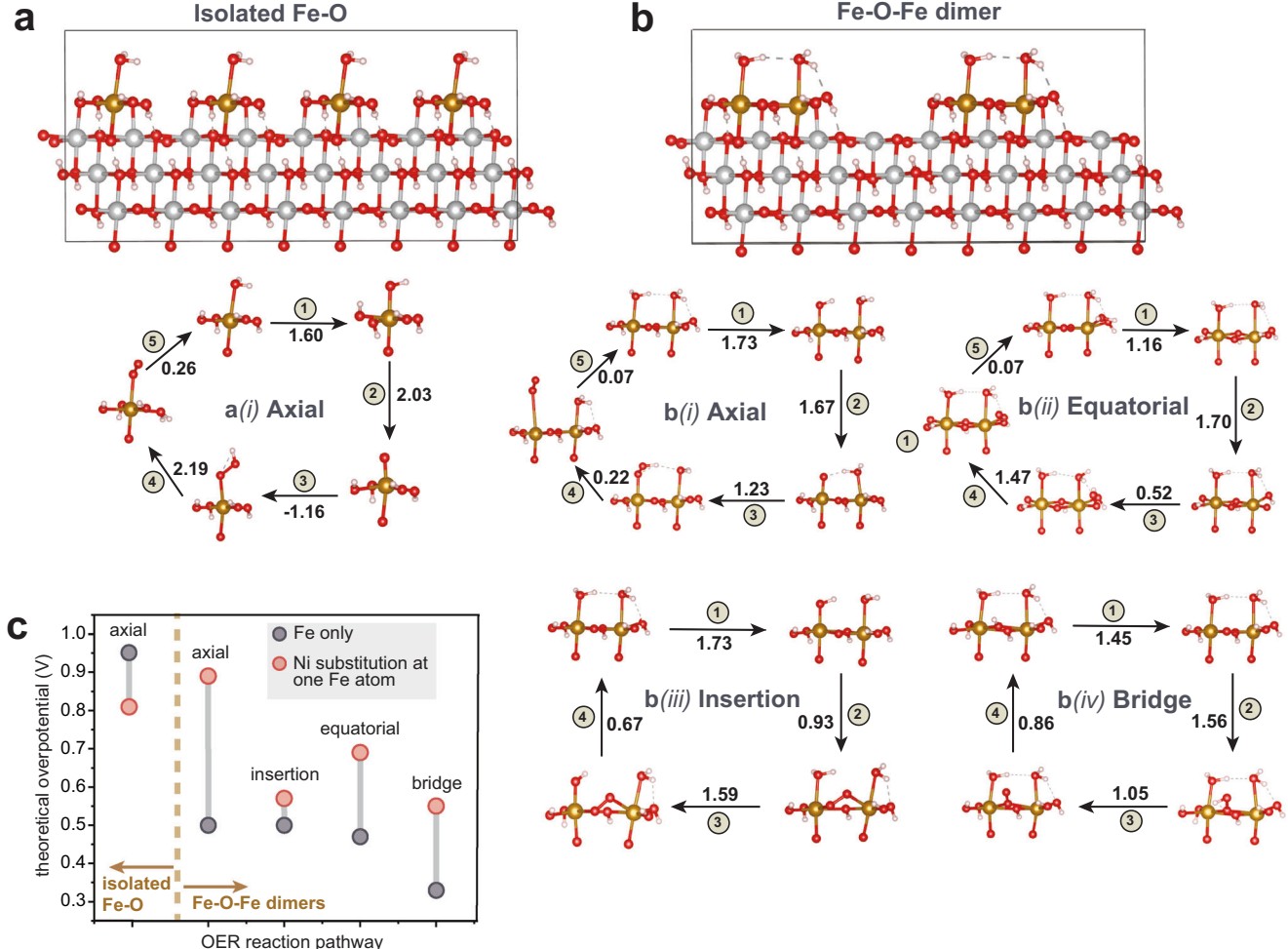

**Fig. 5 | Computational models and OER mechanisms for Fe sites on NiOOH. a, b** Schematic of adsorbed active sites on the NiOOH (0$\bar{1}$5) surface: **a** the isolated "Fe-O" case and **b** dimer "Fe-O-Fe" case, along with the mechanisms chosen for investigation. Values along the reaction pathways are the theoretical overpotential in units of eV for each step. **c** Summary of the total theoretical overpotentials for each depicted pathway including the effect on the overpotential of a single Ni substitution at an Fe in the isolated Fe-O monomer or Fe-O-Fe dimer. Models are shown at a larger scale in Supplementary Fig. 34.

Consequently, the *bridge* mechanism exhibits the lowest overpotential of all mechanisms studied for the Fe-O-Fe dimers with $\eta_{th}$ of 0.33 eV.

In sum, we studied the incorporation and OER-activation by foreign electrolyte ions into electrodeposited $NiO_xH_y$ and $CoO_xH_y$ films and discovered that under controlled oxidative conditions the incorporation can be limited to surface sites. In the case of Fe on $NiO_xH_y$, we used this approach to demonstrate both intrinsically high OER activity (at optimal surface-Fe loading), and a new fundamental picture emphasizing cooperative effects between multiple Fe sites that share oxidative charge. The computations show that new low overpotential pathways for OER are possible through synergistic interaction between multiple Fe species and host-metal atoms whereby oxidative charge can be favorably delocalized and stabilized. The substitution of one Fe by Ni in a model dimer increases the overpotential from 0.33 to 0.55 eV, likely because only one of the three atoms changes its oxidation state, implying that Fe-O-Fe motifs play a key role in charge delocalization. A similar investigation, but for a $CoO_xH_y$ host could explain the discrepancy between $TOF_{Fe}$ of Fe:$NiO_xH_y$ and Fe:$CoO_xH_y$ at fixed Fe loading, namely, $CoO_xH_y$ may not be as effective at sharing oxidative charge with surface Fe-O-Fe. This insight was accomplished through careful electrochemical and analytical techniques−in large part the tools of material characterization (SEM/TEM/XAS) fail to provide these insights because of the disordered nature of the $NiO_xH_y$ support and low Fe loadings associated with the highly active surface $FeO_x$ species.

This work thus provides insight into the exceptional OER activity of Fe-based mixed-metal oxyhydroxide catalysts from which design principles emerge that are important for advanced alkaline water electrolyzers (operating in hot concentrated basic electrolyte), alkaline membrane electrolyzers (that use a solid-ionomer electrolyte and pure water or dilute soluble electrolyte salts), and for broad classes of photoelectrochemical systems. Maximal OER activity is not only a function of the Fe content in $NiO_xH_y$ and $CoO_xH_y$[13] but also exactly where that Fe is located and how it interacts. Because the Fe sites are dynamic, depositing from electrolyte but also simultaneously dissolving[10–13], advanced electrolyzers should engineer the electrolyte "impurity" levels and/or the local dissolution/re-deposition environment such that the highly active cooperative Fe sites can be continually maintained during operation. These sites should be optimally supported on $NiO_xH_y$-based oxyhydroxides surfaces that provide electrical interconnection and nanostructure, but for operational stability and during, e.g., potential changes with on/off cycles, structurally more-robust $CoO_xH_y$ (or other oxides) can be alloyed[65] with the host $NiO_xH_y$ and supported on stable porous-transport layers.

## Methods
### Solution preparation
Stock solutions of 0.1 M Ni(NO$_3$)$_2$·6H$_2$O (Sigma Aldrich, 99.999% trace metal basis) and 0.1 M Co(NO$_3$)$_2$·6H$_2$O (BeanTown Chemical, ≥

99.999% trace metal basis) were separately prepared in 18.2 MΩ·cm water for the electrodeposition of $NiO_xH_y$ and $CoO_xH_y$ films. For the co-deposition of $Ni(Fe)O_xH_y$ and $Co(Fe)O_xH_y$, $FeCl_2 \cdot 4H_2O$ (ACROS Organics, 99+%) was freshly dissolved in $N_2$-purged 18.2 MΩ·cm water to 0.1 M. Then 0.1 M $Ni(NO_3)_2 \cdot 6H_2O$ or $Co(NO_3)_2 \cdot 6H_2O$ was purged with $N_2$ for ~20 min prior to the addition of 0.1 M aq. $FeCl_2 \cdot 4H_2O$. Stock solutions of semiconductor grade 1.0 M KOH (Sigma Aldrich, 99.99% trace metal basis) and further-purified "Fe-free" 1.0 M KOH were used for electrochemical measurements. The electrolyte purification was conducted as reported previously[8]. Briefly, about 2 g of $Ni(NO_3) \cdot 6H_2O$ was dissolved in ~4 mL of 18.2 MΩ·cm water in a 50 mL centrifuge tube and high purity $Ni(OH)_2$ precipitated by rapid addition of ~20 mL of 1 M semiconductor grade KOH. The green $Ni(OH)_2$ precipitate was washed by 2 rounds of addition of 20 mL of water/ 2 mL of 1 M KOH solution. After the final wash, the supernatant was removed by centrifugation and the tube was filled with semiconductor grade 1 M KOH. The tube was shaken vigorously to redisperse the precipitate and allowed to sit overnight during which time the $Ni(OH)_2$ absorbs Fe impurities. All electrolyte used was pH 13.88 ± 0.06. The Fe-free KOH was recovered by centrifuging and then filtered using a 0.1 μm polyethylene sulfone (PES) syringe filter to remove residual $Ni(OH)_2$ particulates. Stock solutions of 0.1 mM $Fe(NO_3)_3 \cdot 9H_2O$ (Sigma Aldrich, ≥ 99.999% trace metal basis), 0.1 mM $Ni(NO_3)_2 \cdot 6H_2O$ (Sigma Aldrich, 99.999% trace metal basis), and 0.1 mM $Co(NO_3)_2 \cdot 6H_2O$ (BeanTown Chemical, ≥ 99.999% trace-metal basis) were separately prepared for foreign-ion "spiking" experiments. To prevent the precipitation of $Fe^{3+}$, the pH of the 0.1 mM aq. $Fe(NO_3)_3 \cdot 9H_2O$ solution was adjusted to ~2 with $HNO_3$.

## Film preparation
All electrodepositions were performed with a two-electrode configuration using pre-cleaned carbon cloth as the counter electrode. Prior to deposition, a bare substrate was cycled five times in Fe-free 1.0 M KOH to verify its cleanliness and induce hydrophilicity. $Ni(Fe)O_xH_y$ was cathodically electrodeposited at $-0.1\,mA \cdot cm^{-2}$ for 120 s. $Co(Fe)O_xH_y$ was electrodeposited at $-2\,mA \cdot cm^{-2}$ for 8 s. The total metal-ion concentration in the electrodeposition bath was 0.1 M. For co-electrodeposited films, the Fe/Ni and Fe/Co ratios were adjusted by the Fe content in the deposition solution. Pt/Ti (50/10 nm) on glass slides, or Pt/Ti-coated quartz crystal microbalance (QCM) crystals (5 MHz, Stanford Research Systems QCM 200), were used as the substrates for electrodeposition. The deposited composition and molar amount were determined by elemental analysis (see below). The mass loading of Ni in the $NiO_xH_y$ film was ~2.7 μg·cm$^{-2}$. The mass of Co in the film of $CoO_xH_y$ was ~5.0 μg·cm$^{-2}$. The film mass loading and change during electrochemistry were determined by the frequency change of the QCM electrode using the Sauerbrey equation ($\Delta f = -C_f \cdot \Delta m$, where $\Delta f$ is the measured frequency change of quartz crystal, $C_f$ is the sensitivity factor with a value of 56.6 Hz cm$^2$·μg$^{-1}$, and $\Delta m$ is the mass change per unit area, μg·cm$^{-2}$).

## Electrochemical characterization
Electrochemical measurements were made with a potentiostat (Bio-Logic SP300 or SP200) using a typical three-electrode setup. Different Hg/HgO reference electrodes (CH Instruments) were used for Fe-free and Fe-containing measurements. The standard electrode potential of the 1 M KOH Hg/HgO reference ($E^0_{Hg/HgO}$) was calibrated to be 0.094–0.099 V vs. NHE[41]. All recorded potentials were converted to the reversible hydrogen electrode (RHE) scale by $E_{RHE} = E_{Hg/HgO} + E^0_{Hg/HgO} + 0.059 \cdot pH$, where $E_{Hg/HgO}$ is the recorded electrode potential vs. Hg/HgO. A Pt coil was used as the counter electrode and pre-cleaned by aqua regia and 18.2 MΩ·cm water. Because alkaline media etches glass leaching Fe into the electrolyte, all measurements were made using inexpensive plastic (poly-methylpentene) cells. The glass section of the substrate was covered by epoxy (Loctite, EA 9460) and hot glue (a commonly

available hot-melt polymer adhesive). The overpotential ($\eta$) was calculated by the equation: $\eta = E_{RHE} - 1.23\,V - iR_u$, where $R_u$ is uncompensated series resistance. $R_u$ was determined by equating $R_u$ to the minimum impedance between 10 kHz and 1 MHz, where the phase angle was closest to zero[41]. The double-layer capacitance ($C_{DL}$) of the metal-oxyhydroxide films was determined by fitting the results of potentio-electrochemical impedance spectroscopy (PEIS) measurements in a potential region where the electrocatalyst films are conductive[66] over the frequency range of 0.1 Hz to 1 MHz (Supplementary Figs. 21 and 22). Experimental details for the XAS measurements are provided in the Supplementary Information.

## Metal-ion-spiking
In CA tests, catalyst films were first polarized in purified Fe-free semi-conductor-grade 1.0 M KOH electrolyte for 3–5 min. Then 715 μL of 0.1 mM metal ion aq. solution was added dropwise into the 40 mL of KOH electrolyte to reach a concentration of added metal of 0.1 ppm (wt. Fe/wt. $H_2O$). The low concentration of 0.1 ppm was chosen to mitigate effects of insoluble $Fe(OH)_3$ colloids or particles that form at higher Fe concentrations and obfuscate the effects of the absorbed Fe species. We estimate from solubility constants[67] that the solubility of $Fe^{3+}$ is about 35 ppb in 1 M KOH and have historically used 0.1 ppm spiking amounts with no observation of precipitation. In voltammetry studies, the catalyst films were first measured in purified Fe-free or semiconductor grade 1.0 M KOH electrolyte for 2–4 cycles to get a stable electrochemical response. Then 715 μL of 0.1 mM metal-ion solution was added dropwise into the 40 mL KOH electrolyte. Magnetic stirring was used throughout the measurement to promote the transport of the added metal ions and prevent bubble accumulation. After measurements, the electrode was removed and rinsed in 18.2 MΩ·cm water.

## Loading analysis and turnover frequency (TOF) calculation
Film mass loading of each element was determined by ICP-MS (iCAP-RQ Qnova Series, Thermo Fisher Scientific). Calibration curves were prepared from third-party-certified reference solutions of the analyte of interest. Electrodes were immersed in 2 mL of 10 v/v% $HNO_3$ for at least 24 h to dissolve the catalyst films. The $HNO_3$ solution was then diluted with 2 mL 18.2 MΩ·cm water for ICP-MS analysis. To verify the full dissolution of catalysts into $HNO_3$ solution, the substrates were rinsed with 18.2 MΩ·cm water and cycled in Fe-free KOH. The lack of Ni or Co redox peaks and similar OER activity to bare Pt/Ti substrate indicated the full dissolution of catalyst films[41]. To compare the intrinsic activity of Fe sites in different matrices, the $TOF_{Fe}$ based on the total mass of Fe in the film was calculated at a constant overpotential[41]:

$$TOF_{Fe} = \frac{current/4F}{(mol\ of\ Fe\ sites)} \tag{1}$$

where F is faraday's constant. The number of Fe sites was determined by ICP-MS. The OER current was recorded from real-time $iR_u$ compensated chronoamperometry tests.

## DFT calculations
Vienna ab initio Simulation Package (VASP)[68–70] was used to perform the spin-polarized DFT calculations with the DFT+U formalism of Duradev et al.[68,69,71]. For the effective modeling of DFT+U, U-J terms of 5.5 and 4 for Ni[72–75] and Fe[76,77] were used, respectively. All calculations were performed using the Perdew–Burke–Ernzerhof (PBE)[78] exchange-correlation functional of the generalized gradient approximation (GGA). The projected augmented wave (PAW) potentials[70,79] include the contribution of core electrons of each atom. An energy cut-off of 600 eV with k-point mesh of 1×1×1 was used for the entire calculation in accordance with the values reported in previous work (65,66). The structures were minimized with energy and force convergence criteria

of $10^{-4}$ eV and $-0.03$ eV·Å$^{-1}$, respectively. Gaussian smearing[80] was used with symmetry imposition for all calculations. The geometries were relaxed with a conjugate gradient algorithm[81].

The overpotential ($\eta$) of each reaction pathway is defined as the minimum potential required to make all reaction steps exothermic. Based on the calculations of the Gibbs free energy of each reaction step, the theoretical overpotential was calculated as follows[82]:

$$\eta = \max(\Delta G_1, \Delta G_2, \Delta G_3, \Delta G_4, \Delta G_5) - V_{OER} \qquad (2)$$

where $\Delta G_i$ is reaction energies of each step and $V_{OER}$ is the equilibrium potential of water oxidation and its reported value is 1.23 eV. The potential energies obtained from the density functional theory calculations are converted to Gibbs free energies as detailed in the Supplementary Information. The overpotential changes were analyzed in terms of changes in the atomic oxidation states of Fe and Ni of each of the reaction intermediate. The oxidation state of Ni and Fe was inferred from the atomic magnetization observed in the output file. For Ni +2, +3 and +4 oxidation states are related to two, one and no singly occupied orbitals, respectively. For Fe +2, +3, +4 and +5 oxidation states are related to four, five, four and three singly occupied orbitals, respectively. The reduction in magnetization values observed results in increase of formal oxidation state. The Gibbs free energy reported here includes the zero-point energy correction.

## Data availability
The electrochemical, materials, and calculation data generated in this study have been deposited in the Figshare database at https://doi.org/10.6084/m9.figshare.24197487.

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

## Acknowledgements

This work was supported by the National Science Foundation Chemical Catalysis Program, Award # 1955106. The ICP-MS instrument was funded by an NSF MRI Award #2117614. Y.O. and L.L. acknowledge support from the China Scholars Council. Dr. Michaela Burke Stevens is thanked for fruitful discussion. This research was supported by the Nancy and Stephen Grand Technion Energy Program (GTEP), by the Israeli Science Foundation (grant no. 880/20), and by a grant from the Israeli Ministry of Science and Technology. This article is based upon work from COST Action 18234, supported by COST (European Cooperation in Science and Technology). We thank the Helmholtz-Zentrum Berlin für Materialien und Energie for the allocation of synchrotron radiation beamtime and for the use of the EMIL infrastructure. The authors acknowledge Dr. Michael Haumann and Dr. Götz Schuck for their support at the beamline. The XAS experiments were financially supported by funds allocated to Prof. Holger Dau (Freie Univ. Berlin) by the Bundesministerium für Bildung und Forschung (BMBF, 05K19KE1, OPERANDO-XAS) and by the Deutsche Forschungsgemeinschaft (DFG, German Research Foundation) under Germany's Excellence Strategy—EXC 2008—390540038—UniSysCat. This project received funding from the European Research Council (ERC) under the European Union's Horizon 2020 Research and Innovation Programme under grant agreement No. 804092.

## Author contributions

Y.O., S.W.B., and J.L.F. conceived the research. S.W.B. directed the research project. Y.O. collected most of the experimental data, with assistance from L.L. and L.T.; B.S and S.B. completed the computational studies, directed by M.C.T.; Y.O., L.T., B.S., M.C.T. and S.W.B. analyzed the data and wrote the manuscript. N.S., J.V., J.M.S., D.A., and M.R. collected and analyzed the XAS data along with L.T.

## Competing interests

The authors declare no competing interests.
