## [Peer Review File · Nature Communications]

REVIEWERS' COMMENTS

Reviewer #4 (Remarks to the Author):

Overall, the authors have provided enough rationale on why certain measurements are not feasible at the moment. I would recommend the acceptance. However, I would also add that some published papers have problems with EXAFS analysis, but there are also published papers where the EXAFS analyses are of high quality because the raw data quality is high. I encourage the authors to consider a model catalyst platform and more consideration of the dynamic nature of the catalytic active sites in future studies.

Response to Reviewer Comments

Reviewer #4: Overall, the authors have provided enough rationale on why certain measurements are not feasible at the moment. I would recommend the acceptance. However, I would also add that some published papers have problems with EXAFS analysis, but there are also published papers where the EXAFS analyses are of high quality because the raw data quality is high. I encourage the authors to consider a model catalyst platform and more consideration of the dynamic nature of the catalytic active sites in future studies.

Thank you for your feedback. We are very interested in analysis of high-quality EXAFS data on this model system because we think this is ultimately the best way to learn more about surface and bulk Fe local environments. We have already submitted a proposal to the Advanced Light Source where we aim to use a new technique which has higher sensitivity than the end station we utilized at BESSY II for this study.